# Delivery of RNA to the Blood-Brain Barrier Endothelium Using Cationic Bicelles

**DOI:** 10.3390/pharmaceutics15082086

**Published:** 2023-08-04

**Authors:** Joan Cheng, Lushan Wang, Vineetha Guttha, Greg Haugstad, Karunya K. Kandimalla

**Affiliations:** 1Department of Pharmaceutics, Brain Barriers Research Center, College of Pharmacy, University of Minnesota, Minneapolis, MN 55455, USA; cheng842@umn.edu (J.C.); wang7886@umn.edu (L.W.); gutth001@umn.edu (V.G.); 2The Characterization Facility, College of Science and Engineering, University of Minnesota, Minneapolis, MN 55455, USA; haugs001@umn.edu

**Keywords:** cationic bicelles, lipid nanodiscs, RNA, Alzheimer’s disease, blood-brain barrier, inflammation, drug delivery, nitrogen-to-phosphate ratio

## Abstract

Blood-brain barrier (BBB) dysfunction is prevalent in Alzheimer’s disease and other neurological disorders. Restoring normal BBB function through RNA therapy is a potential avenue for addressing cerebrovascular changes in these disorders that may lead to cognitive decline. Although lipid nanoparticles have been traditionally used as drug carriers for RNA, bicelles have been emerging as a better alternative because of their higher cellular uptake and superior transfection capabilities. Cationic bicelles composed of DPPC/DC_7_PC/DOTAP at molar ratios of 63.8/25.0/11.2 were evaluated for the delivery of RNA in polarized hCMEC/D3 monolayers, a widely used BBB cell culture model. RNA-bicelle complexes were formed at five N/P ratios (1:1 to 5:1) by a thin-film hydration method. The RNA-bicelle complexes at N/P ratios of 3:1 and 4:1 exhibited optimal particle characteristics for cellular delivery. The cellular uptake of cationic bicelles laced with 1 mol% DiI-C18 was confirmed by flow cytometry and confocal microscopy. The ability of cationic bicelles (N/P ratio 4:1) to transfect polarized hCMEC/D3 with FITC-labeled control siRNA was tested vis-a-vis commercially available Lipofectamine RNAiMAX. These studies demonstrated the higher transfection efficiency and greater potential of cationic bicelles for RNA delivery to the BBB endothelium.

## 1. Introduction

Cerebrovascular disease is observed in 80–90% of patients with Alzheimer’s disease (AD) and has been identified to intensify cognitive decline [1]. The cerebrovascular pathology is primarily characterized by the loss of blood-brain barrier (BBB) integrity and function, which can be observed in both Alzheimer’s patients [2] and transgenic mice models of AD [3]. Activation of inflammatory pathways [4] and disruption of endothelial insulin signaling [5] are believed to engender BBB dysfunction. Therefore, therapies that target the BBB endothelium to alleviate inflammation and rectify insulin signaling are needed to ameliorate BBB dysfunction.

Ribonucleic acid (RNA)-based technologies provide promising approaches to address BBB dysfunction because of their ability to target and modulate relevant genes in a given pathway which are otherwise not amenable to pharmacological agents [6]. However, the bioavailability of RNA-based therapeutics is limited because of their poor permeability across cell membranes and susceptibility to degradation by RNAses in the plasma. To overcome these limitations, the RNA therapeutics are commonly encapsulated in a carrier, among which lipid nanodiscs (LNDs), and have been gaining recognition as superior to conventional lipid nanoparticles (LNPs) for vascular delivery [7]. Discoidal particles exhibit greater cellular uptake than spherical particles for two reasons. One, they naturally orient themselves along the planar edge against the vessel wall during blood flow, maximizing their interaction with the vascular endothelium [8]. Two, they have a lower free energy of membrane curving, leading to faster internalization [9,10].

LNDs, also referred to as bicelles, are spontaneously formed from mixtures of long-chain and short-chain lipids. Some of these lipids may be cationic to enable favorable electrostatic interactions with nucleic acids and cellular membranes. Yang et al. observed that decreasing the molar ratio of long-chain lipid (1,2-dipalmitoyl-sn-glycero-3-phosphocholine, DPPC) to short-chain lipid (1,2-diheptanoyl-sn-glycero-3-phosphocholine, DC_7_PC) from 9:1 to 3:1 resulted in the transformation of the multilamellar liposomes into disc-shaped bicelles [11]. In a subsequent study, they adjusted the composition to include cationic lipids such as 1,2-dioleoyl-3-trimethylammonium-propane (DOTAP) for DNA packing [12]. Cationic bicelles were formed from 8.5 mM DPPC, 3.33 mM DC_7_PC, and 1.5 mM DOTAP (i.e., a molar ratio of 63.8/25.0/11.2). The molar ratio of long-chain lipids to short-chain lipids in these cationic bicelles was maintained at 3:1, and the concentration of DOTAP was isoelectric to the concentration of DNA packed into the bicelles.

The most interesting aspect of this composition is the choice of short-chain lipid. Although most bicelle compositions use 1,2-dihexanoyl-sn-glycero-3-phosphocholine (DHPC), the use of DC_7_PC is more advantageous because of its lower critical micellular concentration (CMC). At concentrations below the CMC of the short-chain lipid, the equilibrium is shifted towards maintaining the concentration of free lipid in the solution [13,14], which results in the depletion of short-chain lipids from the edges and the transformation of the nano-disc into a vesicle. Therefore, a lower CMC for the short-chain lipid allows the bicelles to be diluted to a greater degree. The CMC of DC_7_PC is 1.4 mM while the CMC of DHPC is 15 mM [15], thus bicelles composed of DC_7_PC are deemed to be more stable. Despite their potential for in vivo delivery, DPPC/DC_7_PC/DOTAP bicelles have not been further explored. Therefore, the current study aims to fine-tune and evaluate the potential of these cationic bicelles for RNA delivery to the BBB endothelium.

## 2. Materials and Methods

### 2.1. Materials

1,2-dipalmitoyl-sn-glycero-3-phosphocholine (DPPC), 1,2-diheptanoyl-sn-glycero-3-phosphocholine (DC_7_PC), and 1,2-dioleoyl-3-trimethylammonium-propane (DOTAP) were purchased as chloroform solutions from Avanti Polar Lipids (Alabaster, AL, USA). 1,1′-Dioctadecyl-3,3,3′,3′-Tetramethylindodicarbocyanine-5,5′-Disulfonic Acid (DiI-C18) was purchased from Frontier Chemical (Boulder, CO, USA). Torula yeast RNA, Triton X-100, heparin sodium salt (grade 1-A), and Dulbecco’s Modified Eagle Medium (DMEM) were purchased from Sigma–Aldrich (St. Louis, MO, USA). Control siRNA conjugated to FITC (sc-36869) was purchased from Santa Cruz Biotechnology (Dallas, TX, USA). Lipofectamine RNAiMAX was purchased from ThermoFisher Scientific (Waltham, MN, USA). Human basic fibroblast growth factor (bFGF) was purchased from PeproTech (Cranbury, NJ, USA). Fetal bovine serum (FBS) was purchased from Atlanta Biologicals (Flowery Branch, GA, USA). Dulbecco’s phosphate buffered saline (DPBS, 1X) was purchased by Mediatech (Manassas, VA, USA). All other chemicals were analytical grade.

### 2.2. RNA-Bicelle Complexes Preparation

A mixture of DPPC/DC_7_PC/DOTAP at a molar ratio of 63.8/25.0/11.2 (or DPPC/DC_7_PC/DOTAP/DiI-C18 at a molar ratio of 62.8/25.0/11.2/1.0 for fluorescent studies) was prepared in a round-bottom flask (RBF). Chloroform was removed through rotary evaporation with a Heidolph Laborota 4001 (Schwabach, Germany), and thin lipid film was dried overnight in a desiccator. The film was then hydrated with an aqueous solution of RNA from torula yeast, the concentration of which depended on the N/P ratio.

The RBF was briefly vortexed between two 5 min intervals of sonication in a water bath at room temperature (RT). Free RNA and lipids were removed through centrifugal filtration using Amicon Ultra-2 100K centrifugal filters (Millipore; Burlington, MA, USA). The samples were centrifuged at 2000 rpm (385× *g*) for 6 min at room temperature using an Allegra X-30R Centrifuge, after which they were reverse spun at 500 rpm (24× *g*) for 3 min at room temperature to recover the complexes. Aggregates were removed by centrifuging at 10,000 rpm (18,447× *g*) for 6 min at 4 °C using an accuSpin Micro 17R centrifuge. The supernatant was recovered and the pellet was discarded.

### 2.3. Dynamic Light Scattering (DLS) and Zeta Measurements

Samples were loaded into universal ‘dip’ cells and analyzed using a Malvern Zetasizer Nano ZS instrument. The average hydrodynamic radius (z-average), polydispersity index (Pdi), and zeta potential (ZP) were measured at 25 °C.

### 2.4. Transmission Electron Microscopy (TEM)

Copper grids supported with carbon film were cleaned and hydrophilized by a PELCO easiGlow™ Glow Discharge Cleaning System. Samples were dropped onto the grid and left for 1 min before the excess solution was adsorbed with a filter paper. Negative staining was applied with uranyl acetate (2% wt) for 30 s and excess liquid was absorbed with a filter paper. TEM images were then obtained at 120 kV using a Tecnai G2 Spirit BioTWIN transmission electron microscope.

### 2.5. Atomic Force Microscopy (AFM)

Samples were cast from solution onto freshly cleaved muscovite mica and dried overnight in a desiccator. AFM was performed in ambient (~40% RH) air with a Keysight 5500 operated in AC mode attractive regime [16]. The XYZ scanner was calibrated to within 5% error on a commercial calibration grating. Images were collected with a single silicon probe (NuNano SCOUT 150, nominal spring constant 18 N/m) at 0.5–0.75 lines per second. The cantilever was vibrated at a resonant free amplitude of 9 nm and setpoint amplitude of 7–8 nm.

Post-processing (Gwyddion v.2.58) included (1) initial plane fit/subtraction to remove arbitrary tilt; (2) height-threshold masking of protrusions followed by line-by-line median height alignment of pixel rows; (3) 3rd-order polynomial surface subtraction to remove nonlinearity due to Z scanner and cross-coupling [16]. Statistical analysis invoked Z thresholding to demarcate protrusions followed by compilation of height and diameter histograms.

### 2.6. Ribogreen Assay 

RNA concentrations were measured using a Quant-it™ RiboGreen RNA Assay Kit (Invitrogen; Waltham, MA, USA). Samples were prepared in Molecular Probes^®^ 96-well microplates (Invitrogen) and the assay was performed according to the protocol provided by the manufacturer. Fluorescence intensity was measured (excitation = 485 nm, emission = 530 nm) using a SpectraMax ix3 microplate reader (Molecular Devices; San Jose, CA, USA). Data were analyzed using the SoftMax program.

#### 2.6.1. Encapsulation Efficiency (EE%)

Samples were diluted 25-fold in TE buffer, and then 1000-fold in a digest solution. The digest solution consisted of 10% Triton X-100, 5 mg/mL of heparin in TE buffer, and TE buffer at a volume ratio of 1.0/0.5/98.5. Samples were incubated in the digest solution for 15 min at room temperature. RNA concentrations were determined through Ribogreen assay. Two high-range standard curves were prepared: one in TE buffer and one in digest solution. Reverse pipetting was performed during the addition of the Quant-it™ RiboGreen RNA Reagent to avoid the formation of bubbles by the digest solution.

The EE% was calculated based on Equation (1) as previously described [17], where free RNA concentration is the sample concentration in TE buffer multiplied 25-fold and total RNA concentration is the sample concentration in the digest solution multiplied 2500-fold.
(1)$$EE\%=\left(\frac{Total\;RNA\;Concentration-Free\;RNA\;Concentration}{Total\;RNA\;Concentration}\right)*100\%$$

#### 2.6.2. Drug Release

Samples were loaded into Spectra-Por^®^ Float-A-Lyzer^®^ G2 membranes (MWCO 100 kDa) (Spectrum Laboratories; New Brunswick, NJ, USA) and placed in beakers of DPBS under constant stirring at room temperature. Aliquots of the dialysate were taken at several time points and replenished with fresh DPBS. 

RNA concentrations were measured through Ribogreen assay and expressed as percentages of the expected maximum RNA concentration if RNA were to be completely released from the bicelles. This value was determined by measuring the concentration of the sample after diluting 25-fold in TE buffer and then 500-fold in digest solution. Then, the resultant value was multiplied by the total volume of sample added to the Float-A-Lyzer^®^ G2 membranes and divided by the total volume of DPBS.

### 2.7. Cell Culture Treatments 

Human cerebral microvascular endothelial cell line (hCMEC/D3) was gifted by P-O. Couraud (Institute Cochin; Paris, France). The in vitro BBB model, constituting polarized hCMEC/D3 monolayers, was formed by seeding the cells on plates coated with collagen, and culturing them at 37 °C under 5% CO2 in endothelial cell basal medium (Sigma–Aldrich, St. Louis, MO, USA) supplemented with 5% *v*/*v* FBS.

#### 2.7.1. Cellular Uptake

Polarized hCMEC/D3 monolayers were incubated with 100 µL of bicelles laced with 1 mol% DiI-C18 at 0, 6.67, or 13.33 mM concentrations in 1.5 mL of DMEM at 37 °C for 1 h. Following the incubation, the cells were washed with ice-cold DPBS and processed further. For flow cytometry, the cells were trypsinized, quenched with FBS, and resuspended in 250 μL of 4% *w*/*v* paraformaldehyde (PFA). For confocal microscopy, the cells were fixed with 300 μL of 4% *w*/*v* PFA. After 15 min, the cells were washed twice with ice-cold DPBS and mounted with ProLong Diamond mounting medium containing DAPI nuclear stain (Invitrogen).

#### 2.7.2. Cytotoxicity

Cytotoxicity was assessed by Live/Dead Viability/Toxicity Kit (Thermo Scientific; Waltham, MA, USA). Polarized hCMEC/D3 monolayers were incubated with 75 µL of bicelles (6.67 mM) prepared at four N/P ratios (bicelles alone, 2:1, 3:1, or 4:1) in 1.5 mL of DMEM media at 37 °C for 1 h. The cells were then washed twice with ice-cold DPBS, trypsinized, quenched with FBS, and resuspended in 1 mL of DPBS. Each sample had 2 μL of 50 μM Calcein AM working solution and 4 μL of 2 mM Ethidium homodimer-1 (EthD-1) stock. The samples were vortexed and incubated for 15–20 min at room temperature in the dark.

The stained cells were then analyzed using flow cytometry. The cells were separated into four groups based on their intracellular fluorescence: dead cells in Q1 (Calcein AM−, EthD-1+), compromised cells exhibiting both signals in Q2 (Calcein AM+, EthD-1+), live cells in Q3 (Calcein AM+, EthD-1−), and cell debris in Q4 (Calcein AM−, EthD-1−).

#### 2.7.3. Transfection Efficiency (TE%)

Equimolar amounts of FITC-labeled siRNA (5 pmol) were mixed with bicelles at N/P ratio of 4:1 or with Lipofectamine RNAiMAX, according to the protocol provided by the manufacturer. Polarized hCMEC/D3 monolayers were then incubated with 50 µL of siRNA and bicelles/Lipofectamine in 500 µL of DMEM media at 37 °C for 3 h or 6 h. After the incubation period, cells were washed with ice-cold DPBS and fixed with 300 μL of 4% *w*/*v* PFA. After 15 min, the cells were washed again twice with ice-cold DPBS and mounted with ProLong Diamond mounting medium containing the nuclear stain DAPI. The cells were then imaged through wide-field microscopy, and the TE% was determined based on the mean fluorescence of FITC normalized by the mean fluorescence of DAPI.

### 2.8. Flow Cytometry

Fluorescence signal was captured using an LSR-II Fortessa flow cytometer equipped with 561 nm laser and appropriate band-pass filters. Fluorescence from DiI-C18 was detected by a 585/15 band-pass filter. Green fluorescence from Calcein AM was detected by a 585/15 band-pass filter and red fluorescence from EthD-1 was detected by a 610/20 band-pass filter. The data were analyzed using FlowJo software v10.9.0 (BD Biosciences).

### 2.9. Confocal Microscopy

The confocal micrographs were obtained using a Zeiss LSM 780 laser confocal microscope equipped with a C-Apochromat 40×/1.2 W objective. The DAPI and DiI-C18 were visualized with Zeiss filter sets 49 (excitation 320–380 nm, emission 420–470 nm) and 15 (excitation 600–650 nm, emission 670–720), respectively.

### 2.10. Wide-Field Microscopy

Images were taken using an AxioObserver Z1 inverted microscope equipped with a 100×/1.46NA Alpha Plan-Apo objective and QuantEM 512SC EMCCD camera (Photometrics). The DAPI and FITC were visualized with Zeiss filter sets 49 (excitation 320–380 nm, emission 420–470 nm) and 38HE (excitation 450–490 nm, emission 500–550 nm), respectively. Image analysis was conducted using ImageJ software v1.53s (National Institutes of Health).

## 3. Results

### 3.1. Cationic Bicelle Characterization

The size, Pdi, and ZP of the bicelles were measured through DLS analysis (Table 1). The initial size and ZP were around 210–220 nm and 75–80 mV, respectively. These values did not significantly change after 24 h, which indicated low particle aggregation most likely due to electrostatic repulsion among particles.

TEM images of the bicelles showed their discoidal shape (Figure 1). Flat and circular shapes corresponding to the planar region (i.e., bicelle viewed from above), whereas long and cylindrical shapes indicated the edge regions (i.e., bicelle viewed from the side).

Particle height profiles depicted in the AFM image further corroborated the discoidal shape of the bicelles (Figure 2) with diameters ranging from 60 to 180 nm (average of 138 ± 30 nm) and heights ranging from 4 to 8 nm (average of 6.0 ± 1.2 nm).

### 3.2. RNA-Bicelle Characterization (Effect of N/P Ratio)

The size, ZP, and Pdi of the RNA-bicelle complexes after centrifugal filtration were measured through DLS analysis (Figure 3). Complexation was evident by the proportional decrease in ZP of RNA-bicelle complexes compared to cationic bicelles. The ZP of 1:1 complexes was reduced to near zero (4–6 mV) while the ZP of 2:1 complexes was reduced by half (35–40 mV). In general, higher N/P ratio was associated with a smaller particle size and greater ZP. However, no significant change in Pdi was observed with change in the N/P ratio.

Aggregation was significantly higher for 1:1 complexes, but reduced substantially for 2:1, 3:1, 4:1, or 5:1 complexes, where the population of large aggregates in the suspension contributed to less than 5% of the total peak area. Thus, these large aggregates were removed through centrifugation with minimal loss of particles within the desired size range and without substantially affecting the overall size distribution. For the 3:1 complexes and 4:1 complexes, the z-averages were reduced from ~213 nm to ~141 nm and from ~219 nm to ~127 nm, respectively, upon centrifugation (Figure 4).

A direct correlation between the N/P ratio and EE% was observed (Figure 5). The 1:1 complexes had the lowest EE% (~85.2%) while the 5:1 complexes had the highest EE% (~96.1%). All RNA-bicelle complexes exhibited excellent EE% values. However, due their tendency to aggregate and lower EE%, the 1:1 complexes were excluded from future studies.

Bilayered plate-like structures similar to those reported for DNA-bicelle complexes [12] were observed in the TEM images of RNA-bicelle complexes obtained after centrifugal filtration (Figure 6). However, these structures were not seen in the 5:1 complexes, where unencapsulated bicelles were largely observed, possibly due to the low drug loading. Thus, the 5:1 complexes were excluded from the future studies.

TEM images of RNA-bicelle complexes obtained after centrifugation (Figure 7) showed the same bilayered plates, thus indicating that they are not substantially disrupted by centrifugal filtration. Moreover, their structures appear to be relatively stable as they can still be observed up to 11 days in suspension.

The Live/Dead Cell Viability/Toxicity assay indicated that cationic bicelles were cytotoxic, whereas RNA-bicelle complexes at N/P ratios of 2:1, 3:1, and 4:1 did not show appreciable cell death (Figure 8). Cells treated with cationic bicelles reduced cell viability (~58.9%) compared to the control, whereas the cells treated with 2:1, 3:1, or 4:1 complexes did not significantly increase cell death compared to the control.

The release profiles of the RNA-bicelle complexes were determined through dialysis (Figure 9). Prior to this, (1) bicelles alone were confirmed as a negative control and (2) RNA was observed to generally follow the Higuchi model of diffusion with a plateau around 8 h. A controlled release profile was observed for all three complexes, with release plateau being reached after 48 h. Moreover, an inverse correlation between the N/P ratio and the release plateau was observed. Because the low drug release may hinder its therapeutic efficacy, the 2:1 complexes were excluded from future studies.

### 3.3. Cell Culture Treatments

The concentration-dependent cellular uptake of cationic bicelles laced with 1 mol% DiI-C18 was observed by flow cytometry (Figure 10). A significant shift in fluorescence was observed in cells treated with bicelles compared to untreated cells, indicating that cationic bicelles were successfully internalized by the cells. However, there is only a modest difference between cells treated with 13.33 mM bicelles and with 6.67 mM bicelles.

Confocal images were obtained from the control cells and the cells treated with 13.33 mM of cationic bicelles (Figure 11). The DiI-C18 fluorescence (orange) was observed in the cells treated with cationic bicelles but not in the untreated cells. Without specific staining, it was difficult to discern whether the fluorescence was co-localized with the cellular membrane or with endo/lysosomes.

Cells were incubated with siRNA-Lipofectamine complexes or siRNA-bicelle complexes at an N/P ratio of 4:1. Widefield images were then taken after incubation periods of 3 h and 6 h (Figure 12). For siRNA-Lipofectamine complexes, the TE% was ~57.9% following 6 h incubation and ~77.3% for siRNA-bicelle complexes after 3 h incubation (Table 2). However, the TE% of siRNA-bicelle complexes decreased to 35.6% after 6 h. Therefore, an optimal transfection time must also be considered.

## 4. Discussion

Traditionally, therapeutics against neurodegenerative diseases such as AD are mostly small molecules or monoclonal antibodies. The continual failure of these therapeutic agents in clinical trials has motivated researchers to shift towards novel targets [18]. One such target is to address BBB dysfunction, which has been observed in 80–90% of AD patients and widely recognized to intensify cognitive decline [1]. Molecular pathways driving BBB dysfunction can be modulated using RNA-based therapeutics rather than small molecules or antibodies, as a wider range of molecules involved in the pathways can be simultaneously targeted [6]. Moreover, they can be paired with pharmacological and non-pharmacological treatments to achieve synergistic effects.

However, naked RNA has poor permeability across cell membranes and is susceptible to degradation by RNAses, making its delivery to the site of action difficult. While chemical modification of RNA could improve its permeability and stability, the method is costly and may potentially disrupt its therapeutic activity [19]. Entrapment of RNA in a targeted delivery system is therefore regarded as the most effective method. Initially, viral vectors were used as RNA carriers; however, their safety concerns related to immunogenicity and cost-of-production lead researchers to explore other delivery systems [20]. Currently, nanoparticles fabricated from lipids or polymers are being widely considered for targeting RNA. Of the two, lipids are preferred due to ease of scale-up and avoidance of organic solvents during preparation.

Conventional LNPs are spherical, but alternative shapes are being explored. Bicelles are advantageous in particular, as they offer many of the same benefits of LNPs but their discoidal shape enables greater cellular uptake. In the bloodstream, they are able to tumble and orient themselves along the planar region against the cell wall, allowing for a greater surface area of contact and firmer adhesion with the endothelial cells lining the vessel wall [8]. Once they are adhered to the endothelial membrane, they are internalized more rapidly because of their lower free energy of membrane curving [5,6]. Cationic bicelles composed of DPPC/DC_7_PC/DOTAP at molar ratios of 63.8/25.0/11.2 were previously shown to be excellent carriers for DNA [12]; therefore, they are explored as carriers for RNA in this study.

Cationic bicelles were successfully formed through thin-film hydration. Their hydrodynamic size was around 200 nm and zeta potential around 75 mV. Complexation between the bicelles and RNA resulted in a proportional decrease in the zeta potential, which was found to be dependent on the N/P ratio. For in vivo delivery to the BBB endothelium, particle sizes around 100–200 nm are preferred. Sizes smaller than 100 nm have poor drug loading and may experience burst release of the cargo, whereas sizes larger than 200 nm may not be internalized by receptor-mediated endocytosis [21]. Furthermore, larger particles are more rapidly cleared from the systemic circulation by the reticuloendothelial system [21]. RNA-bicelle complexes were confirmed to be within this size range after the removal of small percentage of large aggregates (<5%) from the suspension.

The structure of RNA-bicelle complexes was also similar to that of DNA-bicelle complexes [12], and appeared as bilayered plates (i.e., bicelles and nucleic acids stacked in alternating layers along the planar regions of the bicelles). This suggests that similar complexes can be assembled between the bicelles and any linear nucleic acid. The removal of larger aggregates through centrifugation did not appear to substantially disrupt RNA-bicelle complexes. The structures also remained intact after 11 days in solution, suggesting that the bicelle structures are fairly stable.

RNA-bicelle complexes were prepared at five different N/P ratios (1:1, 2:1, 3:1, 4:1, and 5:1) to determine the effects of the N/P ratio on the biophysical and physiological characteristics suitable for in vivo delivery. A higher N/P ratio corresponded to a lower drug loading. The RNA-bicelle complexes with lower N/P ratios had a greater tendency to aggregate due to the reduced electrostatic repulsion between the particles [13,22]. A lower N/P ratio also resulted in a lower encapsulation efficiency, most likely due to the greater amount of RNA that is needed to be encapsulated. This correlation between N/P ratio and EE% has been previously reported [17,23].

Although cationic lipids allow for favorable electrostatic interactions with nucleic acids and the cell membrane, they can also be cytotoxic because of their ability to disrupt cell integrity [24] and/or the activity of enzymes such as protein kinase C [25] and Na^+^/K^+^-ATPase [26]). As expected, cells treated with cationic bicelles exhibited significantly greater cell death than untreated cells. However, cells treated with RNA-bicelle complexes did not exhibit significant cell death. This is most likely due to the reduction in the positive charge density, which has been shown to correlate to cytotoxicity, through electrostatic interaction with nucleic acids [24].

Furthermore, slower drug release rates were observed for lower N/P ratios. This was similarly observed in lipoplexes formed between DOTAP and siRNA, where a decrease in N/P ratio (10:1 to 5:1 to 2:1) corresponded to a slower release rate over the course of 48 h, as well as a decrease in *d*-spacing (i.e., the thickness of the bilayer with siRNA sandwiched between) [27]. Thus, it is believed that by lowering the N/P ratio, the particle structure becomes more compressed and RNA is held more tightly, resulting in a slower release rate.

Based on these characterization studies, RNA-bicelle complexes at N/P ratios of 3:1 and 4:1 were determined to exhibit higher cellular uptake and best particle characteristics, including optimal drug loading, particle sizes within the range of 100–200 nm, encapsulation efficiency >90.6%, lower cytotoxicity, and >26% of total drug released over 48 h.

An in vitro BBB cell culture model (polarized hCMEC/D3 monolayers) was treated with cationic bicelles laced with 1mol% DiI-C18. Flow cytometry data showed a substantial increase in intracellular fluorescence in cells treated with cationic bicelles compared to untreated cells, which is indicative of cellular internalization of particles. However, no significant change was observed between cells treated with 13.33 mM and 6.67 mM. Bicelles are believed to be taken up by endocytosis, and low endocytic activity in the BBB endothelium [28] may lead to saturation at higher bicelle concentrations. However, DOTAP is fusogenic [29,30] and the cationic bicelles may also undergo fusion with the cell membrane to directly release their cargo into the cell. Confocal images showed fluorescence from DiI-C18 within the cells treated with cationic bicelles, but it was difficult to determine whether the fluorescence is co-localized with the cell membrane or entrapped in the intracellular vesicles without endo/lysosomal staining. The exact mechanism of cellular uptake will be determined in future studies.

As the “gold standard” for in vitro gene delivery, Lipofectamine RNAiMAX was selected to represent the conventional LNP against which the cationic bicelles were compared. Bicelles demonstrated better in vitro transfection compared to Lipofectamine as they were able to transfect a greater percentage of cells in a shorter period of time. Therefore, cationic bicelles are expected to demonstrate improved in vivo transfection compared to conventional LNPs. While aspects such as PEGylation or targeting ligands will need to be developed further prior to in vivo studies, this study has established the potential of cationic bicelles for the delivery of RNA to the BBB endothelium.

Although the results of this study were largely discussed in the context of BBB dysfunction prevalent in AD, they are applicable to several neurological disorders such as stroke, multiple sclerosis, meningitis, or traumatic brain injury, where BBB dysfunction is central to disease pathology [31]. Abnormal expression of various non-coding RNAs in these diseases can contribute to BBB dysfunction. For example, miR-212 and miR-132 are overexpressed in brain microvascular endothelial cells (BMEC) after traumatic brain injury, and decrease the expression of tight junction-associated proteins [32]. Thus, cationic bicelles are expected to carry RNA-therapeutics to address these disorders.

## Figures and Tables

**Figure 1 pharmaceutics-15-02086-f001:**
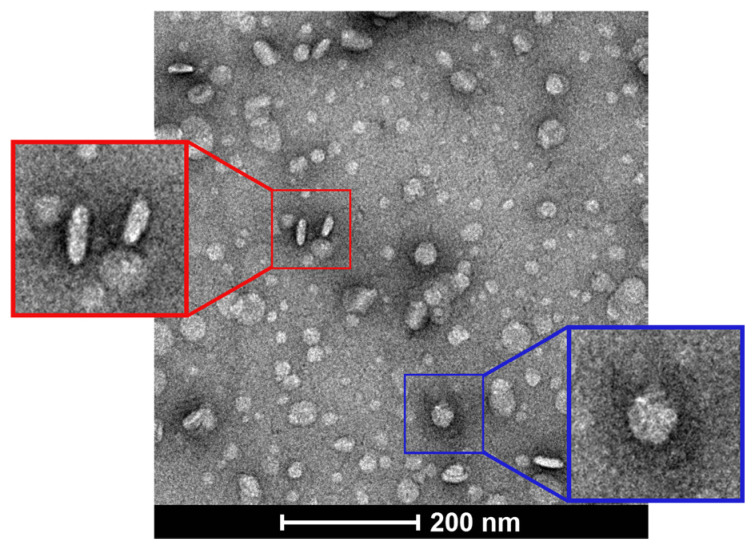
TEM images of cationic bicelles [DPPC/DC_7_PC/DOTAP, molar ratio 63.8/25.0/11.2], negatively stained with 2% uranyl acetate. The bicelle outlined by a blue square indicates the planar view (top) and the bicelle outlined by a red square indicates the edge view (side).

**Figure 2 pharmaceutics-15-02086-f002:**
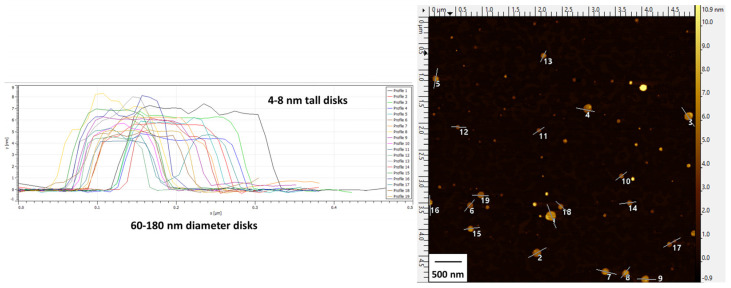
AFM images and height profiles of cationic bicelles [DPPC/DC_7_PC/DOTAP, molar ratio 63.8/25.0/11.2]. Height profiles (**left**) are designated by enumerated lines in the image (**right**).

**Figure 3 pharmaceutics-15-02086-f003:**
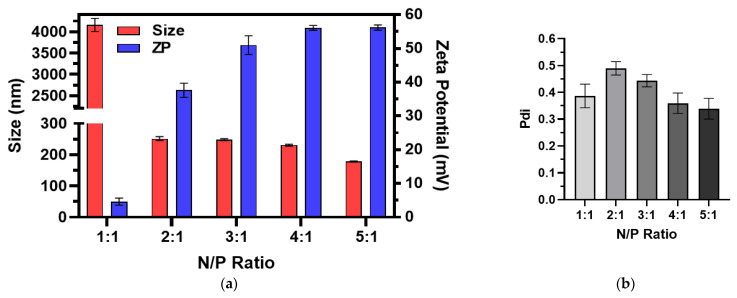
(**a**) Size, zeta potential (ZP), and (**b**) polydispersity indices (Pdi) of RNA-bicelle complexes at varying N/P ratios as determined by DLS (n = 3). Data presented as mean ± SD.

**Figure 4 pharmaceutics-15-02086-f004:**
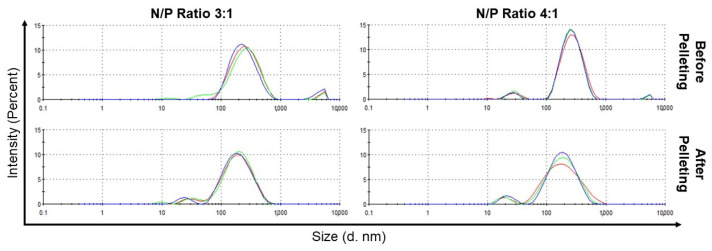
Size distribution, based on intensity as determined by DLS, of RNA-bicelle complexes with N/P ratios of 3:1 and 4:1 before and after centrifugation. Colored lines indicate separate measurements (n = 3).

**Figure 5 pharmaceutics-15-02086-f005:**
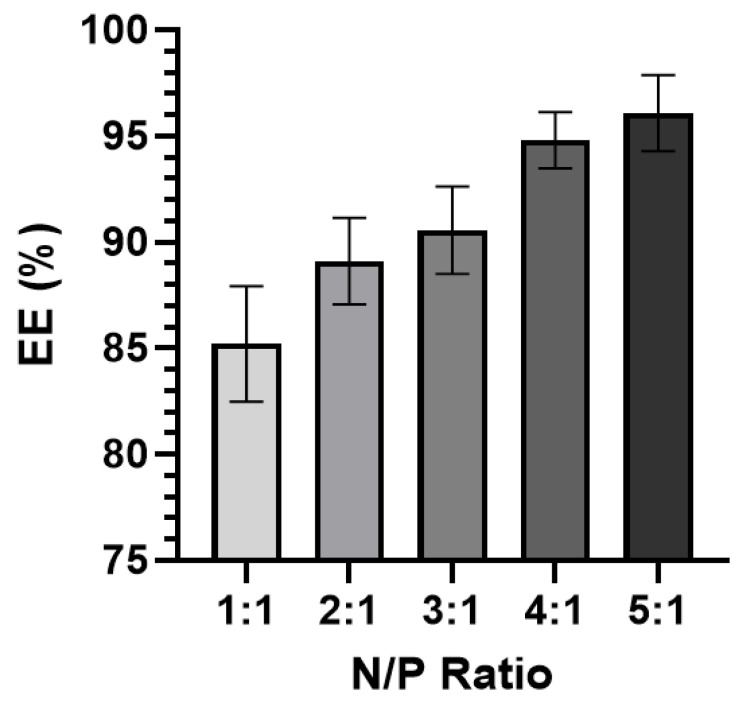
Encapsulation efficiency (EE%) of RNA-bicelle complexes at varying N/P ratios (n ≥ 5). Data presented as mean ± SD.

**Figure 6 pharmaceutics-15-02086-f006:**
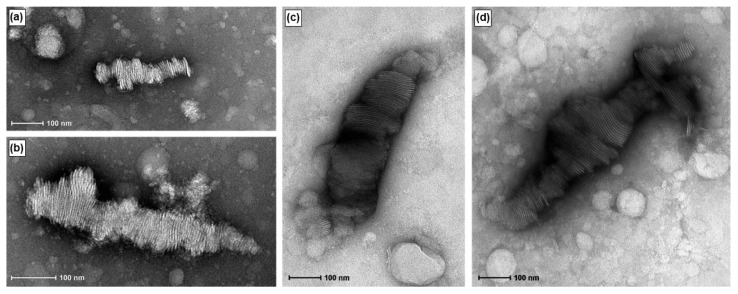
TEM images of RNA-bicelle complexes prior to removal of large aggregates at N/P ratios of (**a**,**b**) 2:1, (**c**) 3:1, and (**d**) 4:1.

**Figure 7 pharmaceutics-15-02086-f007:**
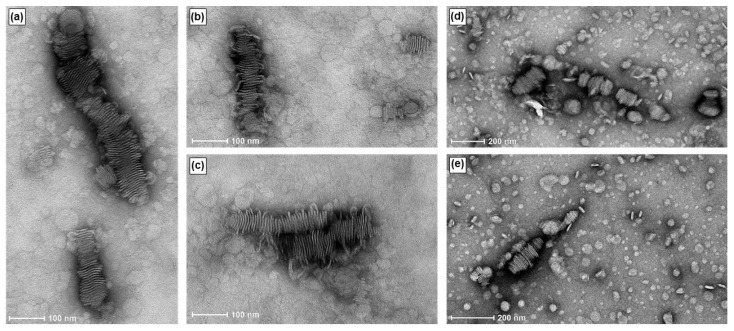
TEM images of RNA-bicelle complexes after removal of large aggregates at an N/P ratio of 3:1. RNA-bicelle complexes were in solution for (**a**–**c**) 5 days or (**d**,**e**) 11 days.

**Figure 8 pharmaceutics-15-02086-f008:**
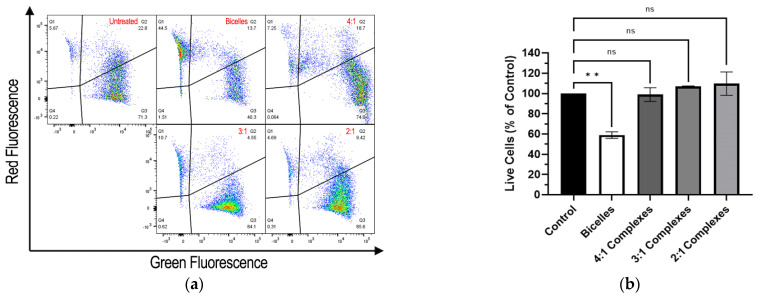
(**a**) Representative flow cytometry of Live/Dead Cell Viability/Toxicity assay and (**b**) percentage of live hCMEC/D3 cells relative to the control after exposure to bicelles, 4:1 complexes, 3:1 complexes, or 2:1 complexes for 1 h (n = 2). Mean ± SD; ** *p* < 0.01, one-way ANOVA followed by Tukey’s post hoc test.

**Figure 9 pharmaceutics-15-02086-f009:**
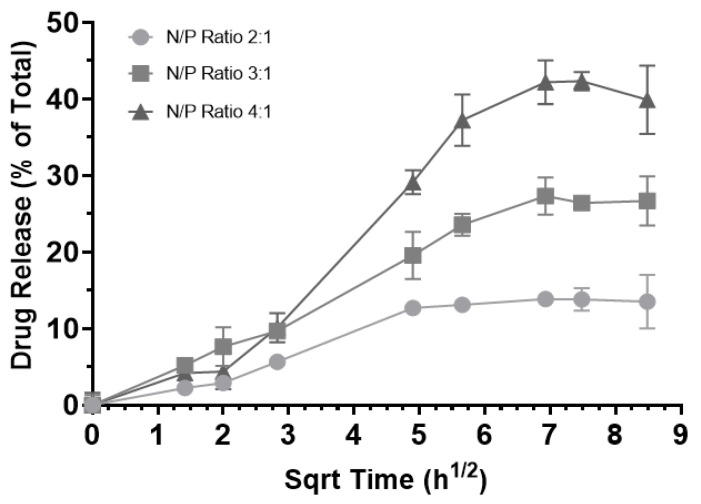
Release of encapsulated RNA from RNA-bicelle complexes at N/P ratios of 2:1 (circle), 3:1 (square), and 4:1 (triangle) over the course of 3 days. Drug release is expressed as a percentage of the total expected concentration, assuming 100% release (n = 3). Data presented as mean ± SD.

**Figure 10 pharmaceutics-15-02086-f010:**
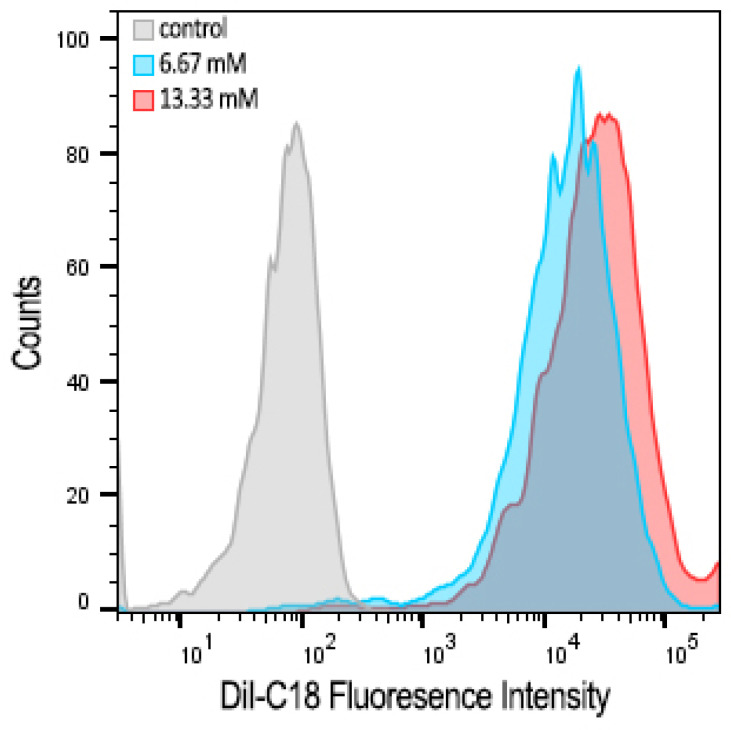
Flow cytometry histograms showing concentration-dependent cellular uptake in hCMEC/D3 monolayers after 1 h incubation with 0 (white), 6.67 (blue), or 13.33 (red) mM of cationic bicelles with 1 mol% DiI-C18.

**Figure 11 pharmaceutics-15-02086-f011:**
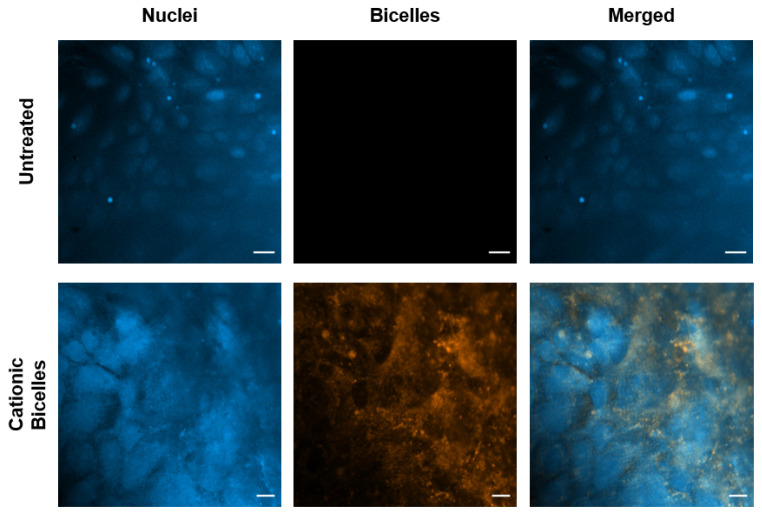
Confocal images of polarized hCMEC/D3 monolayers after 1 h incubation with 0 mM or 13.33 mM of cationic bicelles. Nuclei were stained with DAPI and bicelles were laced with 1 mol% DiI-C18. Scale bar for untreated cells is 20 µm and for treated cells is 10 µm.

**Figure 12 pharmaceutics-15-02086-f012:**
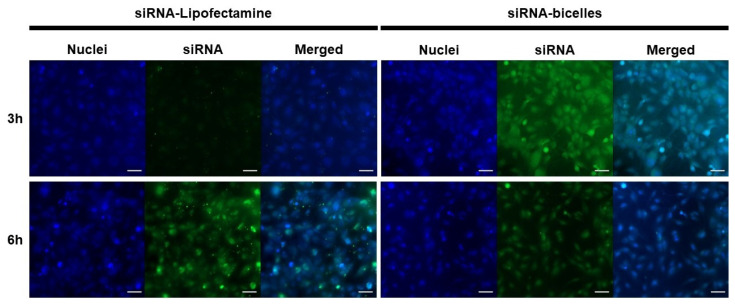
Fluorescence micrographs of polarized hCMEC/D3 monolayers after treatment with siRNA-Lipofectamine complexes or siRNA-bicelle complexes (N/P ratio of 4:1) at 3 h and 6 h. Nuclei were stained with DAPI (blue) and siRNA was labeled with FITC (green). Scale bar is 50 µm.

**Table 1 pharmaceutics-15-02086-t001:** Size, polydispersity index (Pdi), and zeta potential (ZP) of cationic bicelles [DPPC/DC_7_PC/DOTAP, molar ratio of 63.8/25.0/11.2] after 0 h and 24 h determined by dynamic light scattering (DLS) technique (n = 3). Data presented as mean ± SD.

	0 h	24 h
**Size (nm)**	218.7 ± 7.28	204.5 ± 7.28
**Pdi**	0.361 ± 0.025	0.352 ± 0.046
**ZP (mV)**	77.4 ± 2.49	70.2 ± 6.78

**Table 2 pharmaceutics-15-02086-t002:** The TE% of siRNA-Lipofectamine complexes and siRNA-bicelle complexes (N/P ratio 4:1) in polarized hCMEC/D3 monolayers after incubation for 3 h or 6 h (n = 3). Data presented as mean ± SD.

Incubation Time (h)	TE% siRNA-Lipofectamine	TE% siRNA-Bicelle (N/P 4:1)
3	13.1 ± 5.04	77.3 ± 4.71
6	57.9 ± 5.89	35.6 ± 3.10

## Data Availability

The data presented in this study are available on request from the corresponding author. The data are not publicly available due to privacy.

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
