# Peer review of "Delivery of RNA to the Blood-Brain Barrier Endothelium Using Cationic Bicelles"

_pharmaceutics, 2023, doi:10.3390/pharmaceutics15082086_

Round 1
Reviewer 1 Report
The study is well-designed and well-conducted. The authors have effectively investigated the use of cationic bicelles for the delivery of RNA in an in vitro blood-brain barrier (BBB) cell culture model. The research methodology appears sound, and the experimental procedures are appropriately described. The manuscript is well-written, and the findings are presented clearly.
However, there are few recommendations related to the manuscript:
1) One minor suggestion for improvement is to clarify in the manuscript why the authors specifically refer to Alzheimer's disease. It would be beneficial to discuss the relevance of RNA delivery to the brain endothelial cells for other neurological and/or neurodegenerative disorders as well.
2) On which basis have the authors selected the specific molar ratios of DPPC/DC7PC/DOTAP 63.8/25.0/11.2 used in the cationic bicelles? Was the selection based on the previously described literature or was there any optimization performed by the authors?
3) It is optionally recommended that the authors consider expanding the discussion to include a comparison with other RNA delivery strategies, highlighting the unique advantages and limitations of cationic bicelles.
4) There are minor typo errors that should be addressed, for example:
There is an extra parenthesis in the title of section 2.3, and in the caption of Fig. 2 there is a parenthesis missing.
5) In Discussion section, page 12, it is recommended that the authors support their discussion mentioning that 'Preliminary studies have demonstrated successful cellular uptake of DiI-C18 laced cationic bicelles by BBB cell culture models in vitro .' with appropriate references from the literature.
Author Response
1) One minor suggestion for improvement is to clarify in the manuscript why the authors specifically refer to Alzheimer's disease. It would be beneficial to discuss the relevance of RNA delivery to the brain endothelial cells for other neurological and/or neurodegenerative disorders as well.
Response: We added a new paragraph on page 12, lines 453 – 463. This paragraph elaborates on how BBB dysfunction occurs in other neurological diseases such as stroke or multiple sclerosis, and how ncRNAs play a role in their development (see reference 29). For example, miR-212/132 expression was observed to be increased in mouse and human brain microvascular endothelial after traumatic brain injury (simulated through hypoxia). miR-212/132 was observed to decrease the expression of various tight junctions, which resulted in a leaky BBB (see reference 30).
2) On which basis have the authors selected the specific molar ratios of DPPC/DC7PC/DOTAP 63.8/25.0/11.2 used in the cationic bicelles? Was the selection based on the previously described literature or was there any optimization performed by the authors?
Response: We added additional information on page 2, line 47 – 58. This paragraph explains the rationale behind the selection of the molar ratios, as previously demonstrated by Yang et al. (see reference 9 and 10). Additionally, we relocated the paragraph explaining the advantages of DC7PC over DHPC from the discussion to the introduction on page 2, line 59 – 70.
3) It is optionally recommended that the authors consider expanding the discussion to include a comparison with other RNA delivery strategies, highlighting the unique advantages and limitations of cationic bicelles.
Response: We added additional information on page 11, lines 371 – 382. This paragraph explains the advantages of bicelles over other techniques (e.g., chemical modification) or carriers (e.g., viral vectors, polymeric nanoparticles) (see reference 17 and 18).
4) There are minor typo errors that should be addressed, for example: there is an extra parenthesis in the title of section 2.3, and in the caption of Fig. 2 there is a parenthesis missing.
Response: Thank you for pointing those out. We have corrected them.
5) In Discussion section, page 12, it is recommended that the authors support their discussion mentioning that 'Preliminary studies have demonstrated successful cellular uptake of DiI-C18 laced cationic bicelles by BBB cell culture models in vitro .' with appropriate references from the literature.
Response: We appreciate the reviewer’s suggestion. “Preliminary studies” referred to our own data in the paper (i.e., Figure 11). We have corrected the wording on page 12, line 435 – 436 to “An in vitro BBB cell culture model (polarized hCMEC/D3 monolayers) was treated with cationic bicelles laced with 1mol% DiI-C18.”
Reviewer 2 Report
The work być Cheng et al., titled: ”Delivery of RNA to the Blood-Brain Barrier Endothelium Using Cationic Bicellest”, is well-constructed and analyzes all technical aspects as well as the potential application of the discussed cationic bicelles. However, it is necessary to address some minor corrections presented below.
Introduction
„…Activation of inflammatory pathways and disruption of endothelial insulin signaling are believed to be responsible for BBB dysfunction[3]…”
In my opinion, the given reference no. 3, does not fully justify the content of the sentence, please add another one
„…Of these therapies, ribonucleic acid (RNA)-based technologies are very promising because of their ability to target and modulate relevant genes in a given pathway which are otherwise not amenable to pharmacological agents…”
Please add references
„…However, the bioavailability of RNA-based therapeutics is limited because of their poor permeability…”
Please clarify the „permeability” of what you mean.
Material and methods
„…Control siRNA conjugated to FITC was purchased from Santa Cruz Biotechnology (TX, USA)…”
Please add the catalog number
„…2.6.1. Encapsulation Efficiency (EE%)…”
Please provide a reference for the used formula.
Discussion
„…Traditionally, AD research has targeted various pathological hallmarks such as amyloid-β proteins or hyperphosphorylated tau with small molecules or monoclonal antibodies. However, continual failure of these therapeutic agents in clinical trials has motivated researchers to target other pathological features in AD brain. One such target is BBB dysfunction, which has been observed in 80-90% of AD patients and widely recognized to intensify cognitive decline. Molecular pathways driving BBB dysfunction can be modulated effectively using RNA-based therapeutics, which can specifically target critical nodes compared to small molecules or antibodies. Moreover, they can be paired with pharmacological and non-pharmacological treatments…”
There are no references in the entire fragment of the text, please complete it.
Author Response
“…Activation of inflammatory pathways and disruption of endothelial insulin signaling are believed to be responsible for BBB dysfunction[3]…”
In my opinion, the given reference no. 3, does not fully justify the content of the sentence, please add another one
Response: We have included another reference (5) on page 1, line 32. In this paper, endothelial insulin signaling was found to play an important role in suppressing inflammation, maintaining healthy BBB function, and promoting Aβ degradation. Thus, disruption of endothelial insulin signaling could result in BBB dysfunction.
This, alongside reference 3, should justify our statement: “Activation of inflammatory pathways[4] and disruption of endothelial insulin signaling[5] are believed to be responsible for BBB dysfunction.”
“…Of these therapies, ribonucleic acid (RNA)-based technologies are very promising because of their ability to target and modulate relevant genes in a given pathway which are otherwise not amenable to pharmacological agents…”
Please add references
Response: We have included a reference (6) on page 1, line 37. This paper reviews the history, concepts, current status, and future potential of RNA therapy. Advantages of RNA therapy over existing small molecule-based or monoclonal antibody-based therapies are highlighted, such as their ability to target nearly all genetic component within the cell (e.g., noncoding RNAs).
“…However, the bioavailability of RNA-based therapeutics is limited because of their poor permeability…”
Please clarify the “permeability” of what you mean.
Response: We use the term “poor permeability” to refer to the restricted transport of RNA across the cell membrane due to its large size and hydrophilic nature. For clarity, we have changed the phrase to “poor permeability across the membrane” on page 1, line 38 – 39.
“…Control siRNA conjugated to FITC was purchased from Santa Cruz Biotechnology (TX, USA)…”
Please add the catalog number
Response: It has been added on page 2, line 79.
“…2.6.1. Encapsulation Efficiency (EE%)…”
Please provide a reference for the used formula.
Response: We have included a reference (15) on page 3, line 163. In this paper, hybrid nanodiscs conjugated with cRGD (targeting ligand) were loaded with siRNA. Encapsulation efficiency was determined through Ribogreen assay and calculated as (Ctotal – Cfree) / Ctotal x 100% where Ctotal denotes the total siRNA concentration in the nanodisc suspension and Cfree denotes the unloaded siRNA concentration. We adopted the same method and formula for this study.
“…Traditionally, AD research has targeted various pathological hallmarks such as amyloid-β proteins or hyperphosphorylated tau with small molecules or monoclonal antibodies. However, continual failure of these therapeutic agents in clinical trials has motivated researchers to target other pathological features in AD brain. One such target is BBB dysfunction, which has been observed in 80-90% of AD patients and widely recognized to intensify cognitive decline. Molecular pathways driving BBB dysfunction can be modulated effectively using RNA-based therapeutics, which can specifically target critical nodes compared to small molecules or antibodies. Moreover, they can be paired with pharmacological and non-pharmacological treatments…”
There are no references in the entire fragment of the text, please complete it.
Response: We have included reference (18) on page 11, line 364, reference (1) on page 11, line 366, and reference (5) on page 11, line 369. Reference 1 discusses the intersection between cerebrovascular disease, cardiovascular disease, and AD. Reference 16 reviews the general landscape of AD drug development, emphasizing Aβ and hyperphosphorylated tau as the most popular target classes and small molecules and monoclonal antibodies as prevalent treatment modalities. The effect of the failure of most AD trials (i.e., the shift in the landscape towards novel targets and therapies) is highlighted as well.